# Dried Blood Spot Sampling to Assess Rifampicin Exposure and Treatment Outcomes among Native and Non-Native Tuberculosis Patients in Paraguay: An Exploratory Study

**DOI:** 10.3390/pharmaceutics15041089

**Published:** 2023-03-29

**Authors:** Samiksha Ghimire, Gladys Molinas, Arturo Battaglia, Nilza Martinez, Luis Gómez Paciello, Sarita Aguirre, Jan-Willem C. Alffenaar, Marieke G. G. Sturkenboom, Cecile Magis-Escurra

**Affiliations:** 1Department of Clinical Pharmacy and Pharmacology, University Medical Center Groningen, University of Groningen, 9712 CP Groningen, The Netherlands; 2Instituto Nacional de Enfermedades Respiratorias y del Ambiente “Juan Max Boettner”, Asuncion 1430, Paraguay; 3Programa Nacional de Control de la Tuberculosis, Asuncion 1430, Paraguay; 4School of Pharmacy, Faculty of Medicine and Health, The University of Sydney, Camperdown, NSW 2006, Australia; 5Westhead Hospital, West Mead, NSW 2145, Australia; 6Sydney Institute of Infectious Diseases, The University of Sydney, Camperdown, NSW 2006, Australia; 7Department of Pulmonary Diseases, Radboud University Medical Center-TB Expert Center Dekkerswald, 6525 GA Nijmegen, The Netherlands

**Keywords:** tuberculosis, rifampicin, pharmacokinetics, dried blood spots, limited-sampling strategy

## Abstract

The aim of this study was to evaluate the difference in drug exposure of rifampicin in native versus non-native Paraguayan populations using dried blood spots (DBS) samples collected utilizing a limited sampling strategy. This was a prospective pharmacokinetic study that enrolled hospitalized tuberculosis (TB) patients from both native and non-native populations receiving oral rifampicin 10 mg/kg once-daily dosing. Steady-state DBS samples were collected at 2, 4, and 6 h after intake of rifampicin. The area under the time concentration curve 0–24 h (AUC_0–24_) was calculated using a Bayesian population PK model. Rifampicin AUC_0–24_ < 38.7 mg*h/L was considered as low. The probability of target attainment (PTA) was calculated using AUC_0–24_/MIC > 271 as a target and estimated MIC values of 0.125 and 0.25 mg/L. In total, 50 patients were included. Native patients (*n* = 30) showed comparable drug exposure to the non-natives (*n* = 20), median AUC_0–24_ 24.7 (17.1–29.5 IQR) and 21.6 (15.0–35.4 IQR) mg*h/L (*p* = 0.66), respectively. Among total patients, only 16% (*n* = 8) had a rifampicin AUC_0–24_ > 38.7 mg*h/L. Furthermore, PTA analysis showed that only 12 (24%) of the patients met a target AUC_0–24_ /MIC ≥ 271, assuming an MIC of 0.125 mg/L, which plummeted to 0% at a wild-type MIC of 0.25 mg/L. We successfully used DBS and limited sampling for the AUC_0–24_ estimation of rifampicin. Currently, our group, the EUSAT-RCS consortium, is preparing a prospective multinational, multicenter phase IIb clinical trial evaluating the safety and efficacy of high-dose rifampicin (35 mg/kg) in adult subjects using the DBS technique for AUC_0–24_ estimation.

## 1. Introduction

Tuberculosis (TB) remains the leading cause of death by a single infectious agent apart from COVID; being at the same time both a preventable and curable disease. Globally, the burden of TB varies greatly, with more than 60% of the newly diagnosed cases concentrated in parts of Asia and Africa [1,2]. Although the incidence of TB in South America (46.2 per 100,000) is relatively low compared to Africa (100–300 per 100,000 inhabitants), TB still remains a serious public health problem. In Paraguay, TB incidences differ significantly among the native (350 per 100,000) and the non-native population (37.6 per 100,000) [2]. Based on limited data, the mortality rates are estimated to vary considerably with 22 deaths in native Indians compared to 4.5 per 100,000 in the rest of the population [2]. As of yet, no evidence is available to explain the reason why indigenous people have such high incidence rates of TB, why there is a higher mortality rate, and why the cure rates and relapse rates are lower. An explanation could be compliance with medication, higher prevalence of toxicity and side effects, or access to health care. Another reason might be low drug concentrations due to genetic differences (explaining differences in metabolism or toxicity) or due to food and drinking habits. None of these reasons have been studied in Paraguay so far.

Rifampicin is a key drug of standard first-line anti-TB treatment. From hollow-fiber studies and an observational cohort study [3,4], it is evident that low rifampicin blood exposures have been associated with treatment failure and the development of acquired drug resistance [5,6]. Pharmacokinetic (PK) variability, in combination with differences in drug susceptibility of *Mycobacterium tuberculosis*, explains why some patients fail the treatment [6] although the correlation with outcomes in humans has not been completely established until now. For rifampicin, the most predictive efficacy parameter is its ratio of area under the 24-h concentration-time curve (AUC_0–24_) and minimum inhibitory concentration (MIC), with a suggested target exposure of AUC_0–24_/MIC ≥ 271 for one log CFU reduction in murine studies [5,7], whereas free C_max_/MIC ≥ 175 is linked to the suppression of acquired drug resistance and the post-antibiotic effect (hollow-fiber study) [5]. However, in the absence of actual MICs, clinicians often utilize an AUC_0–24_ of 38.7 mg*h/L, observed with normal 8–12 mg/kg dosing in humans to make informed dosing decisions [8].

In a systematic review and meta-analysis, Stott et al. found that among 35 included studies (at steady state), 12 studies had an average AUC_0–24_ between 20–30 mg*h/L, whereas, two other studies had an even lower mean AUC_0–24_ of 13–14 mg*h/L [8]. There was a considerable difference in reported AUCs in the different populations (HIV status, TB status, combination therapy, intermittent dosing, diabetes status, and treatment duration) [8]. Furthermore, Stott et al. showed that by taking 38.7 mg*h/L as a mean rifampicin steady-state AUC and the epidemiological cut-off value (ECOFF) MIC of 0.5 mg/L, an AUC/MIC ratio of 77 is achieved, which is far below the optimal AUC_0–24_ /MIC ≥ 271 suggested by Jayaram et al. [7,8]. However, the proposed susceptibility breakpoint MIC based on pharmacokinetic/pharmacodynamic science was much lower at 0.0078 mg/L for the standard rifampicin dose (450–600 mg once daily) [9,10].

Therapeutic drug monitoring (TDM), i.e., drug dose adjustment based on measured blood concentrations, has the potential to optimize exposure to anti-TB drugs but is not widely used in clinical practice probably due to logistic and financial challenges that arise with the conventional venous sampling [11,12,13]. Encouragingly, the use of alternative sampling techniques such as dried blood spots (DBS), combined with limited sampling, has led to practical solutions [14,15,16]. Limited sampling is drawing a limited number (usually two or three) of samples during one dosing interval to calculate AUC_0–24_. DBS sampling, due to its higher stability and easy shipment to central laboratories, has overcome some of the challenges associated with conventional venous sampling [15,16]. Moreover, recent developments in the field of limited sampling strategies have enabled accurate prediction of drug exposure without the need for intensive pharmacokinetic sampling [17,18]. In Paraguayan pediatric TB patients, pharmacokinetic sampling using DBS collection utilizing limited-sampling time points (0, 2, 4, and 8 h post dose) was successful in accurately assessing the pharmacokinetics of rifampicin and pyrazinamide [18]. Although exposure to rifampicin has been extensively studied, data on rifampicin exposure in the South American adult population is lacking.

The primary aim of this study was to evaluate the differences in exposure to rifampicin in native versus non-native Paraguayans using DBS sampling. The hypothesis to study exposure of rifampicin in native vs. non-native Paraguayans stemmed from the high discrepancy in TB incidence and mortality rates among native Paraguayan patients compared to the rest of the population. Apart from the socioeconomic, cultural, educational, and health barriers which might explain the differences in TB incidence and mortality, we were compelled to study if there were any differences in the exposure of the key anti-TB drugs, such as rifampicin, in this population. None of these reasons have been studied in Paraguay so far. Our study is the start of unraveling the differences between subgroups in the population and we decided to start to investigate exposure in the native population and compare it to another, genetically different, study population. Furthermore, it is uncertain if any related genetic differences or other biological features contributing to the differences should be considered.

### 1.1. Patients and Setting

The study was carried out at the hospital ‘Instituto Nacional de Enfermedades Respiratorias y del Ambiente’ (INERAM) in Asunción, Paraguay. The study protocol was approved by the Ethics Committee of Laboratorio Central De Salud Pública—Asunción (No: 56/160415). From July 2015 to October 2018, hospitalized adult TB patients were asked to participate in the study. Signed informed consent was obtained from all patients. New drug-susceptible TB patients, ≥18 years, with pulmonary or extrapulmonary involvement or patients with high clinical suspicion of TB, or starting treatment with first-line TB drugs were eligible for inclusion. Severe renal or hepatic dysfunction at the start of treatment leading to a different TB treatment regimen was excluded. For this study, a sample size of 50 hospitalized patients (30 native and 20 non-natives) was considered. In the Paraguayan national TB program, a database is available based on the Paraguayan TB epidemiology. Native Paraguayans were identified based on their indigenous identity card which also expanded on ethnic and lingual identity. Furthermore, in this database high-risk groups such as prisoners, indigenous people, and others (substance abusers, HIV, and diabetes) are indicated as such. Native Paraguayans belong to one of the 17 ethnic groups in Paraguay. Of the 17 ethnic groups, 13 live in the western region also called the Chaco of Paraguay (arid land, with very high temperatures, sometimes exceeding 50 °C, and their diet is primarily composed of the meat of wild animals). The other four ethnic groups live very close to the capital, Asuncion, and consume a similar diet to that of non-natives. The non-native population consisted of mestizo people who are of mixed Spanish and native descent.

Given the exploratory nature of the study, no formal sample size calculation was performed in this study. As PK data from the native population is lacking, we used an enrolment ratio of 1.5 to have sufficient samples to compensate for unforeseen circumstances. Patients were followed up until the completion of treatment.

### 1.2. TB Diagnosis, Drug Doses, Study Procedures, and Pharmacokinetic Analysis

Tuberculosis patients were verified based on the results from the sputum-smear-positive Ziehl Neelsen microscopy andculture on solid media, chest X-ray, and other molecular tests such as GeneXpert. GeneXpert was used as a confirmatory test for patients with positive sputum smears.

All patients were treated according to the WHO standard TB treatment guidelines and received once daily rifampicin (10 mg/kg, rounded to 450 or 600 mg) along with isoniazid, pyrazinamide, and ethambutol [19]. Comedication and basic demographics were recorded. Biochemical parameters were determined at baseline and on the day of PK sampling, which included hematocrit, serum creatinine, and liver enzymes. Blood samples were collected with a 10 mL syringe after disinfecting the skin with 70% alcohol. The collected blood was divided into two tubes for the collection of plasma and serum samples. The samples were immediately transferred to INERAM’s laboratory for processing. At the Hospital del Indigena, the samples were collected inside the laboratory. Laboratory tests were performed at the INERAM laboratory and the Hospital del Indigena (Asuncion, Paraguay), both using the same automated Beckman equipment (German origin). The equipment used the enzymatic method for transaminases: alkaline phosphatase (ALP), alanine transaminase (ALT), and aspartate aminotransferase (AST) and the colorimetry method for bilirubins. Hemoglobin and hematocrit were analyzed by the flow-cytometry method. The modified Jaffé kinetic method was used to analyze creatinine. The creatinine reference range was 0.7–1.3 mg/dL. The baseline laboratory tests were used to gain insights into the overall health status of the TB patients which included the patient’s liver and kidney function, as well as their glucose metabolism, more as a descriptive approach. Apart from hematocrit, all other baseline parameters mentioned in Table 1 are routinely analyzed in the TB treatment centers every two to four weeks to evaluate if adverse reactions occur. In case of abnormal results, the test is repeated to take care of the adverse effects.

PK sampling was performed after two weeks of treatment (to ensure a steady state), at 2, 4, and 6 h after observed administration of rifampicin [20]. Patients had to remain fasted for at least two hours prior to rifampicin intake until the second DBS sample (4 h) was taken. On the day of sampling, a trained healthcare worker collected capillary blood from a finger prick on special filter paper (Whatman^®^ DMPK type C, Darmstadt, Germany) to produce DBS samples, and the DBS cards were dried at room temperature (20–35 °C) for at least 3 h and subsequently stored in a plastic ziplocked bag with silica enclosed. Contrary to standardized storage conditions, DBS samples were stored at room temperature before shipping them to the Netherlands. Rifampicin and its metabolite deacetyl-rifampicin were found to be stable at an ambient temperature of approximately 25 °C for 2 months, 37 °C for 10 days, and 50 °C for 3 days [21]. The shipping process to the Netherlands was in accordance with one of the stated conditions (37 °C for 10 days). The ambient temperature inside the local site was below 25 °C at all times due to the use of fans and coolers.

Rifampicin drug concentrations from these DBS samples were analyzed at the laboratory of the Department of Clinical Pharmacy and Pharmacology at the University Medical Center Groningen, Groningen, The Netherlands. The DBS samples were measured using a liquid-chromatography tandem mass-spectrometry technique (LC-MS/MS) assay validated for accuracy, precision, linearity, matrix effect, hematocrit effect, spot volume, and stability upon storage Details on the DBS assay of rifampicin have been published previously [21]. The calibration curve of rifampicin showed to be linear with a correlation coefficient (R^2^) of 0.9953. Within run precision (%CV) ranged from 2.1% to 5.4% and between run precision ranged from 2.4% to 7.2%. Accuracy (bias) ranged from −1.1% to 4.0% at all tested QC levels (LLOQ, low, medium, high, and over the curve). Rifampicin was measured from the DBS cards and reported [21]. While reporting drug levels in two different matrixes (plasma and dried blood spots) there could be a systemic difference in rifampicin concentrations measured in plasma vs. DBS. We tested this in our earlier published clinical application study of venous blood and DBS samples for rifampicin [21]. Simple linear regression coefficients and Deming regression equations for rifampicin were R2 = 0.9076 (*n* = 28), y = 0.90x − 0.01 (95% confidence interval (CI) slope: 0.78–1.01, intercept: 0.53–0.51). The clinical application study showed no significant differences between the patient analysis of plasma and DBS for rifampicin [18,21]. Based on these results, a correction factor was deemed not applicable.

To estimate AUC_0–24_ in this population, a validated one-compartmental population pharmacokinetic model with first-order absorption, lag time, and dose adaptation software was used (MW Pharm++, version 1.9.8.243; Mediware, Prague, Czech Republic) [16]. The currently applied three-point limited sampling strategy was validated in an earlier published study by Magis-Escurra et al. [20], where multiple linear regression analysis was conducted to obtain optimal sampling equations predictive of actual AUC_0–24_ for rifampicin. All possible combinations of one to three time points were evaluated. The average adjusted *R*^2^ and mean absolute percentage prediction error (MAPE) for all combinations comprising one, two, or three samples were calculated to investigate the correlation between the predicted AUC_0–24_ using multiple linear regression and calculated AUC_0–24_. The predictive performance of two-sampling time points and one-sampling time point fell short in terms of R^2^ and MAPE [17,20]. Furthermore, a single PK sample at 2 h would not have been adequate for accurately predicting AUC_0–24_, as it has been well established that Cmax cannot be estimated using C2 level [14].

### 1.3. Sputum-Smear Microscopy, Culture, and Statistical Evaluation

Sputum samples were collected from patients for sputum-smear microscopy and culture weeks after that following the WHO guidelines for the treatment of drug-susceptible TB (DS-TB) [19]. The culture was performed in the Löwenstein-Jensen media. All statistical analysis was performed in SPSS, version 23.0 (IBM Corp., Armonk, NY, USA). Categorical data were expressed in frequencies and percentages whereas continuous variables were presented as median and interquartile range. Nonparametric tests were used for the calculation of *p*-values (Mann–Whitney U test).

### 1.4. Treatment Outcomes

Treatment outcomes were reported following the WHO guidelines [19].

## 2. Results

### 2.1. Study Subjects

In this study, 50 hospitalized TB patients were included (length of hospitalization was about one month), of which 41 (82%) patients had positive sputum smears before starting TB treatment. The baseline characteristics of both natives (*n*= 30) and non-natives (*n*= 20) are summarized in Table 1. The median age of the native population was 35 years (26–52 interquartile range, IQR), and the non-native population was 40 years (28–43 IQR). The actual median rifampicin dose was 600 mg (IQR 525–600 mg). The national TB program provides Rifafour, a combination of the four drugs, in the intensive phase. Rifafour is a fixed-dose combination of four active TB drugs: rifampicin 150 mg, isoniazid 75 mg, pyrazinamide 400 mg, and ethambutol HCL 275 mg. Based on the BMI < 18.5 kg/m^2^, 45% (9/20) of the non-native patients were underweight; and 31% (10/29) of the native patients were underweight. All 50 patients completed PK sampling using DBS; samples were taken in an average of a median 14 days (11–19 IQR) after starting treatment except in five patients who were sampled after 30 days of treatment. Of 50 patients, 10 (20%) patients who were lost to follow up after the first month of treatment were equally distributed over two groups, six (20%) from natives (*n* = 30) and four (20%) from the non-natives (*n* = 20). Of the total, one (2%) patient stopped TB treatment because in the diagnostic tests, all the results were negative for TB and the diagnosis became unlikely.

### 2.2. Pharmacokinetics

At 10 mg/kg once daily dosing, the median AUC_0–24_ (*n* = 50) was 23.2 mg*h/L (16.4–32.3 IQR). Native patients showed a comparable rifampicin exposure to the non-natives; the median AUC_0–24_ was 21.6 (15.0–35.4 IQR) and 24.7 (17.1–29.5 IQR), respectively (*p* = 0.66, Mann-Whitney U test, see Table 2).

Of the total (*n* = 50), only eight (16%) patients had a rifampicin reference AUC_0–24_ > 38.7 mg*h/L; six (20%) patients from the native population and two (10%) from the non-natives, indicating that the Paraguayan population (both native and the non-native) are at the lower end of the AUC values that are observed in TB patients. The probability of rifampicin target attainment (PTA) was calculated (Figure 1) and showed PTA (AUC/MIC > 271) in the Y-axis plotted against the assumed MIC of rifampicin ranged from 0.00156 to 1 mg/L in the X-axis. The analysis (*n* = 50) showed that with once-daily dosing of 450–600 mg, only 27% of the native patients and 20% of the patients from the non-native group met the target AUC_0–24_ /MIC ≥ 271 if MIC was assumed to be 0.125 mg/L. Furthermore, at an assumed MIC of 0.25 mg/L, PTA dropped dramatically to 0% in the total population. At present, the ECOFF MIC of rifampicin is set at 0.5 mg/L, whereas PK/PD susceptibility breakpoint is at 0.0625 mg/L [9]. This is reflected in our cohort (see Figure 1), as the PTA dropped from 100% (at MIC of 0.0625 mg/L) to 0% (MIC of 0.25 mg/L). The most prevalent MIC for rifampicin in Paraguay is 0.25 mg/L (*personal communication with the central laboratory in Asuncion, Paraguay*).

### 2.3. Sputum-Smear/Culture Conversion and Treatment Outcomes

The median time to sputum-smear conversion (*n* = 21) was 15 days (15–45 IQR) and culture conversion (*n* = 25) was 15 days (15–30 IQR). In the native population, the median time to sputum-smear conversion (*n* = 6) was 15 days (15–15 IQR), and culture conversion (*n* = 13) was also 15 days (15–15 IQR). In the non-native population, the median time to sputum-smear conversion (*n* = 15) was 15 days (15–45 IQR) and culture conversion (*n* = 12) was 30 days (15–60 IQR). Sputum-smear/culture conversion data at 60 days of the treatment and the final treatment outcomes of the native and non-native populations are shown in Table 2.

## 3. Discussion

This study evaluated the differences in drug exposure to rifampicin between native and non-native Paraguayans using DBS sampling. Rifampicin exposure in Paraguayan TB patients (natives and non-natives) was considered low as only 16% of the patients achieved a reference AUC_0–24_ > 38.7 mg*h/L [8]. There was no difference in the rifampicin doses (10 mg/kg) between the two groups (*p* = 0.49, see Table 2). No significant difference in rifampicin exposure was observed between the native and non-native Paraguayan population (*n* = 6, 20% in the native and *n* = 2, 10% in the non-native group met the AUC target). Both groups had rifampicin levels towards the lower AUC range (median 23.2 mg*h/L and IQR 16.4–32.3) of normal in patients receiving a dose of 10 mg/kg once daily [8]. Of note, there were higher numbers of female participants in the native group compared to the non-native group. The study by Conte et al. found that absorption of rifampicin or the plasma levels of rifampicin at 2 h and 4 h were not affected by sex [22]. Therefore, we assume that the absorption of rifampicin was comparable between males and females in our study.

Remarkably, rapid conversion in both native and non-native populations occurred despite low exposure to rifampicin. This may indicate that the current treatment during the intensive phase of treatment was adequate, which could be attributed to the combined effect of the four drugs. Several factors could have contributed to the rapid conversion. First, the MIC of rifampicin might have been low in these patients (although the most prevalent MIC for rifampicin in Paraguay is reported at 0.25 mg/L). Second, it could be that combination therapy of isoniazid, pyrazinamide, and ethambutol also contributed to the rapid response. Still, the low exposure in relation to sterilization might have resulted in the dormancy of TB bacteria rather than actually killing it as long-term follow-up data is missing, which is concerning. Therefore, our study shows that the currently used dosing of rifampicin 10 mg/kg is likely suboptimal and there is plenty of room for rifampicin dose optimization [8,23]. Furthermore, these were hospitalized patients who are often sicker than the normal TB patient population. Although other reasons for hospitalization in this study included socioeconomic reasons, patients lived far from the treatment center, were poor, or lived alone with no one to take care of them. Additionally, our study showed that pharmacokinetics can be assessed in remote settings using this strategy in adults as well as in children [18].

Our results are consistent with earlier reports on rifampicin levels in adult TB patients [8,24,25]. Seijger et al. evaluated serious adverse events in a cohort of 88 patients with severe presentations such as meningitis TB in one of our TB-expert centers, treated with high-dose rifampicin (up to 32 mg/kg) from the treatment start or after TDM [26]. Encouragingly, no serious adverse events occurred, and AUC_0–24_ showed nonlinear increases at higher dosages [26]. Therefore, along with our study, the available evidence suggests that the time is now right to apply a higher dose of rifampicin in order to save lives, especially in the most difficult-to-cure patients [25]. With higher doses, not only desired AUC will be attained but also culture conversion will be reached earlier which might result in shortening the therapy duration.

This study has limitations. First, for sputum-culture conversion, a hallmark for evaluating the efficacy of the TB-treatment therapy, data were available only in half of the included patients due to several reasons such as some patients could not expectorate sputum samples, others missed the visit, some were sputum-culture negative from the start, and some had extrapulmonary TB. Second, MIC values were not measured and assumed MIC of 0.125 and 0.25 mg/L were used. Third, in our study, 20% of the patients were lost to follow up. These patients are at greater risk for developing drug resistance and further transmitting resistant forms of TB to the community. Fourth, the model was not validated separately for the native Paraguayan tuberculosis patients [16]. However, the original model was built using a heterogeneous population, with respect to ethnicity, weight, and BMI. Next to that, it was developed for the same dosing range. This model has been successfully in use to calculate AUC24 in daily patient care for almost 10 years now. Based on our vast experience with a very large range of different parameters, we are confident that the model can be applied to estimate exposure in the native population. Unfortunately, the exposures of rifampicin in routine care in the TB center are not part of this project, hence we cannot report on them, nor compare them as they are not part of the study approved by the ethics committee. Fifth, we did not perform the genetic analysis searching for variants in the genes SLCO1B1, ABCB1, UGT1A, or PXR which have a role in the extent of gun/hepatic enzyme induction and metabolism [27]. Sixth, this exploratory study was not powered to find any differences resulting between the groups. The heterogeneity is visible in the rifampicin exposure IQR in both groups, which varied between 15.0–35.4 mg*h/L. Based on a post hoc power analysis, future studies would need a total of 95 patients to achieve 80% power (this is with 15% additional subjects included due to the non-normality of the rifampicin exposure). Therefore, this is our recommendation for any future follow-up studies of a similar nature.

Furthermore, we did not study the pharmacokinetics parameters of rifampicin as it remains outside the scope of this study due to several reasons. First, limited sampling time points (2 h, 4 h, and 6 h) are validated to estimate the exposure of rifampicin over 24 h AUC_0–24_ and not other PK parameters like Cmax, Tmax, half-life, and volume of distribution [20]. To perform a full pharmacokinetics study, we need to collect multiple samples at 0 h (pre-dosing), 30 min, 1 h, 2 h, 3, 4, and 6 h postdosing. Second, since AUC_0–24_ remains the first best predictor of efficacy for rifampicin, we were interested in the exposure difference between the two groups in this first prospective cohort study given the pilot character of the project. Assessing full PK parameters including genotyping and treatment-outcome data would be a follow-up study.

Finally, to address the issue of high loss to follow up, national TB treatment programs in Paraguay are currently focusing on formulating and improving health strategies. Inter-sectoral collaboration, sustained financing, and applying new tools and technologies such as TDM or video-based directly-observed treatment will be important to carve the path for the more efficient functioning of TB programs in Paraguay.

### Future Perspectives, Next Steps

Currently, our group, the EUSAT-RCS consortium (https://www.eusattb.net/), is preparing a multinational, multicenter phase IIb clinical trial evaluating the safety and efficacy of high-dose rifampicin (35 mg/kg) in adult subjects with pulmonary or extrapulmonary DS-TB belonging to difficult to treat subgroups (such as patients older than 60 years and/or with significant comorbidities with active tuberculosis) [28]. The results from this scheduled phase II trial will generate solid evidence on if the optimized dose will be feasible in the clinical practice for the whole population.

## 4. Conclusions

In our cohort of 50 Paraguayan TB patients, rifampicin 10 mg/kg dosing resulted in low exposures in both native and non-native patients. Native patients showed comparable exposure to non-natives. Dried blood spot sampling was used successfully to estimate rifampicin exposure and this sampling method seems feasible in resource-limited settings.

## Figures and Tables

**Figure 1 pharmaceutics-15-01089-f001:**
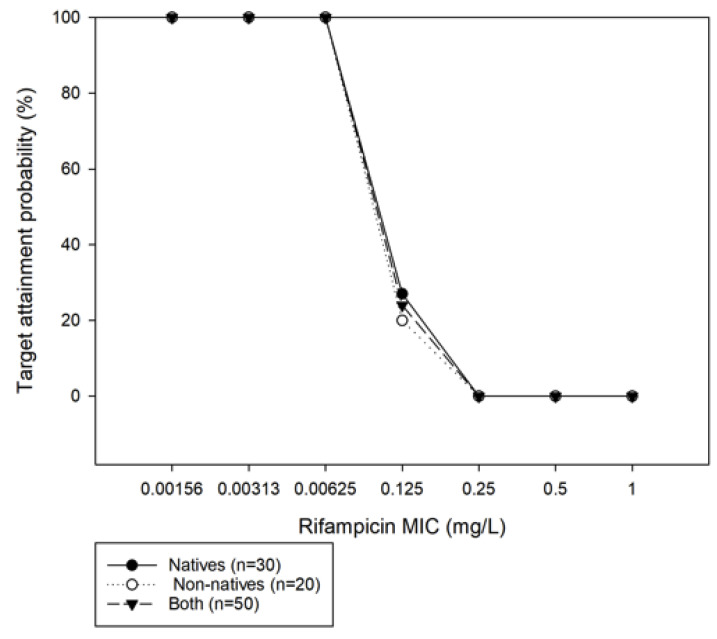
Probability of target attainment (AUC_0–24_/MIC > 271) vs. *minimum inhibitory concentration* (MIC) in patients at assumed rifampicin MIC (ranged from 0.00156 to 1 mg/L)).

**Table 1 pharmaceutics-15-01089-t001:** Baseline characteristics of all included patients.

Patient Characteristics	Total (*n* = 50)	Native (*n* = 30)	Non-Native (*n* = 20)
Demographic data			
Male	33 (66)	16 (53)	17 (85)
Age (years)	35 (26–49)	35 (26–52)	40 (28–43)
Body weight (kg)	52 (47–62)	53 (44–62)	52 (49–63)
Length (cm)	167 (160–173)	164 (151–172)	169 (162–175)
Body Mass Index (kg/m^2^) (*n* = 49)	19.5 (17.9–21.6)	20.2 (18.0–21.8)	18.9 (16.8–21.3)
Underweight (<18.5 kg/m^2^)	19 (38)	10 (33)	9 (45)
Normal (18.5–25.0 kg/m^2^)	31 (62)	20 (67)	11 (55)
Type of TB			
Pulmonary	46 (92)	28 (93)	18 (90)
Extrapulmonary	4 (8)	2 (7)	2 (10)
Comorbidities			
Diabetes	6 (12)	2 (10)	4 (20)
HIV	1 (2)	1 (3.3)	0 (0)
Radiological characteristics (*n* = 47)		(*n* = 27)	(*n* = 20)
Consolidation	36 (77)	23 (85)	13 (65)
Pleural fluid	8 (17)	7 (26)	1 (5)
Cavities	29 (62)	14 (52)	15 (75)
Atelectasis	8 (17)	5 (19)	3 (15)
Baseline Biochemical parameter			
Creatinine (umol/L), normal value 53 to 97.2 umol/L	61.9 (53.09–70.7)	61.9 (53.0–70.7)	61.9 (52.4–70.7)
ALT (U/L), normal value 10–49 U/L	27.5 (16–50.3)	35.0 (20.8–59.3)	20.5 (15.3–36.3)
AST (U/L), lower than 34 U/L	28.5 (19–46.2)	33.5 (23.8–57.5)	28.0 (19.5–34.0)
ALP (U/L), 90–360 U/L	209.0 (58.0–316.75)	233 (161.3–362.3)	232.0 (141.0–309.0)
Haemoglobin (g/dL)	10.65 (9.4–12.3)	10.1 (9.2–11.3)	11.5 (9.6–12.7)
Hematocrit (%)	33.6 (31.0–38.0)	33.1 (31.0–35.7)	36.0 (30.4–39.1)
Baseline Sputum smear (*n* = 49)		*n* = 30	*n* = 19
Positive	41 (83)	24 (80)	17 (89.4)
Negative	8 (16)	6 (20)	2 (10.5)
Baseline culture (*n* = 48)		*n* = 28	*n* = 20
Positive	33 (69)	19 (68)	14 (70)
Negative	15 (31)	9 (32)	6 (30)
GeneXpert MTB/RIF (*n* = 33)		*n* = 21	*n* = 12
Positive	32 (97)	20 (95)	12 (100)
Negative	1 (3)	1 (5)	0 (0)

Data are expressed as median (IQR) or *n* (%).

**Table 2 pharmaceutics-15-01089-t002:** Rifampicin dose, AUC, and treatment outcomes.

	Native (*n* = 30)	Non-Native (*n* = 20)
Rifampicin dose (mg/kg)	10.74 (9.43–11.51)	10.66 (9.47–11.69)
AUC (mg*h/L)	24.7 (17.1–29.5)	21.6 (15.0–35.4)
Primary Treatment Outcomes		
- Sputum conversion (60 days, yes)	16 (94), *n* = 17	10 (91), *n* = 11
- Culture conversion (60 days, yes)	15 (94), *n* = 16	9 (90), *n* = 10
Final treatment outcomes (end of the treatment)		
- Cured	9 (30)	11 (55)
- Treatment completion	10 (34)	4 (20)
- Loss-to-follow up	6 (20)	4 (20)
- Missing data	1 (3)	1 (5)
- Not evaluated	4 (13)	-

Data are expressed as median (IQR) or *n* (%).

## Data Availability

Not applicable.

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
