# Peer review of "Dried Blood Spot Sampling to Assess Rifampicin Exposure and Treatment Outcomes among Native and Non-Native Tuberculosis Patients in Paraguay: An Exploratory Study"

_pharmaceutics, 2023, doi:10.3390/pharmaceutics15041089_

Round 1

Reviewer 1 Report

This is an interesting paper in which investigators endeavored to use dried blood spot (DBS) sampling among adult patients in Paraguay hospitalized with TB to measure rifampicin concentrations in this population. This follows on work by some members of the group in the laboratory (Vu, reference 22) and children (Martial, reference 20).  The authors found that the proportion of patients with rifampicin concentrations in the therapeutic range was low, not unexpectedly.  The paper is a lead-in to an upcoming randomized clinical trial that will use DBS to collect PK samples from patients receiving high-dose versus standard-dose rifampicin.  I have some suggestions for the authors' consideration that I think will be important to support use of this assay for this purpose.  

MAJOR

(1) The authors compare PK of rifampicin in 'natives' versus 'non-natives'. This terminology, to me, is problematic.  I don't know how these terms are defined, if they are acceptable to use, how they were discerned (did patients provide racial and ethnic identity information?), how one might anticipate drug disposition to be different by group and what common characteristics are shared by participants in each group (aren't there multiple ethnic groups among indigenous peoples of Paraguay? what are backgrounds of the non-natives?).  I would suggest removing this component and just describing the use of the DBS among adults with TB in Paraguay. 

(2) DBS samples were collected by finger stick and then put on a Whatman card and kept at room temperature.  Rifampicin was measured from the DBS cards and reported.  In previous studies, a correction factor was required so that DBS rifampicin PK values correlated with plasma values.  How did finger stick DBS compare to venous sampled DBS?  Was a correction factor needed so that DBS values would reflect plasma values?  What was the stability of the assay in the local environment of the study when the sample was kept at room temperature?  Authors should describe the validation steps taken in this new, adult population for capillary DBS measurements of rifampicin.

(3) What were the hemotocrit or hemoglobin values in the study population? This is curiously absent from the baseline characteristics table (Table 1).

MINOR
(1) Could a single PK sample at 2 hours have been adequate, using the prediction software, to predict PK parameters of interest (Cmax or AUC)?

(2) What were the actual doses? That is, who received 450 and who received 600, and what was the distribution?

(3) Rifampicin concentrations were described as below the target for the majority of patients, yet culture conversion was rapid in this patient population.  So the conclusion that higher doses are needed does not follow from the data and so they should be revised.  What is the NAT2 acetylator status distribution in the study population? 

(4) Authors kept DBS samples at room temperatures for up to 2 months prior to shipping to the testing lab.  Were validations done to ensure drug concentrations were not affected by this? 

Author Response

Dear reviewer,

Thank you for the constructive feedback and the detailed review of our manuscript. 

We have thoroughly revised the manuscript based on the queries and addressed all the technical comments and useful suggestions. Please find in attachment a point-by-point response.

Reviewer 2 Report

Review - Dried blood spot sampling to assess Rifampicin exposure and treatment outcomes among native and non-native tuberculosis patients in Paraguay

This paper compares drug exposure of rifampicin between native and non-native Paraguayan tuberculosis patients. For the purpose of therapeutic drug monitoring dried blood spot sampling is used combining with limited sampling strategy. Based on obtained results pharmacokinetic-pharmacodynamic evaluation is also performed.

Major remarks

1.    The major aim of this research was mentioned as “to evaluate the difference in drug exposure of rifampicin in native versus non-native Paraguayan population using dried blood spots (DBS) samples collected utilizing limited sampling strategy” (line: 20-22). The usefulness of this comparison is not explained obviously. It just could be assumed that significant difference in rifampicin exposure is supposed because:

o   Is there a genetic cause, or another biological feature underlying the differences between Paraguayans and non-Paraguayans?

o   Did the authors validate the applied PK model (MW Pharm++, version 1.9.8.243; Mediware, Prague, Czech Republic) for native Paraguayan tuberculosis (TB) patients. This information would be very appropriate for the planned phase IIb clinical trial, however in my opinion this connection is not built into the manuscript.

Please clarify and elaborate on the importance of the comparison which is noted as the main aim of the study. Please incorporate the detailed explanation appropriately into the manuscript.

2.     DBS sample stability is characterized as follow: “Rifampicin and its metabolite deacetyl-rifampicin were found to be stable at ambient temperature of approximately 25 ÌŠC for 2 months, 37 ÌŠC for 10 days and 50 ÌŠC for 3 days [22]” (line: 162-164). Please let the readers know whether the shipping process to the Netherlands is accordance with one of the stated condition set (25 °C – 2 months; 37 °C – 10 days, 50 °C – 3 days). If not, did you use any approach to control stability of DBS samples. Please detail it in the manuscript, if applicable.

3.      Non-native group includes only 3 female patients (3/20, 15%), however in the native group 14 female patients (14/30, 47%) are registered. Please state with references that differences in the rate of sex could not influence the comparison between native and non-native patients group.

4.      Probability target attainment evaluation is not detailed appropriately. Please provide more detail on the software and the parameter settings applied during the analysis.

5.      Line 100-102: “However, in the absence of actual MICs, clinicians often utilize AUC0-24 of 38.7 mg*h/L, observed with normal 8-12 mg/kg dosing in humans to make informed dosing decisions [12]”. Reference 12 (Abulfathi AA, Decloedt EH, Svensson EM, Diacon AH, Donald P, Reuter H. Clinical pharmacokinetics and pharmacodynamics of rifampicin in human tuberculosis. Clin Pharmacokinet. 2019;58(9):1103–29) is an review article and not contain directly the mentioned specific AUC value of rifampicin. Please cite the original article.

6.      Line 110-112: “In contrast, the proposed breakpoint MIC based on pharmacokinetic/pharmacodynamic science was much lower at 0.0078 mg/L for the standard rifampicin dose (450-600 mg once daily) [10]”. It is hard to understand what “in contrast” means in this context. Please rephrase this sentence to be clearer.

7.      Line 186-188: “Depending on the distribution of continuous variables, non-parametric tests were used for calculation of p-values, where applicable (Mann Whitney U test)”. Please specify the cases in which non-parametric, Mann Whitney U test was employed and also detail the cases when parametric comparison could be conducted.

8.      Line 251-253: “This may indicate that the current treatment during the intensive is adequate which could be attributed to the combined effect of four drugs. Several factors could have contributed to the rapid conversion”. “During the intensive” phrase is hard to understand, maybe authors thought the intensive treatment or intensive pharmacotherapy. Please edit this sentence.

Minor remarks

1.      Line 29-31: “In total 50 patients were included. Native patients (n=30) showed comparable drug exposure as the non-natives (n=20), median AUC0-24 24.7 (17.1-29.5 IQR) and 21.6 (15.0-35.4 IQR) mg*h/L (p=0.66).” Missing from the end of the sentence: , respectively.

2.      Line 107-110: “Furthermore, Stott et al. showed that taking 38.7 mg*h/L as a mean rifampicin steady state AUC and the epidemiological cut-off value (ECOFF) MIC of 0.5 mg/L, AUC/MIC ratio of 77 is achieved, which is far below the optimal AUC0-24 /MIC ≥271 suggested by Jayaram et al. [6,8,9] “. Two articles are referenced with specific results , however three citations are set as references. Please correct the references.

3.      Line 221: “(personal communication with the central laboratory in Asuncion, Paraguay)”. Personal communication could be the part of the manuscript as a supplementary materiel, if it is applicable.

4.      Line 297: “DS-TB”. The abbreviation DS appears for the first time in the manuscript. Please edit the sentence.

Author Response

(The authors gave the same response as above.)
